# Antitumor Potential of *Annona muricata* Linn. An Edible and Medicinal Plant in Mexico: In Vitro, In Vivo, and Toxicological Studies

**DOI:** 10.3390/molecules26247675

**Published:** 2021-12-18

**Authors:** Verenice Merlín-Lucas, Rosa María Ordoñez-Razo, Fernando Calzada, Aida Solís, Normand García-Hernández, Elizabeth Barbosa, Miguel Valdés

**Affiliations:** 1Doctorado en Ciencias Biológicas y de la Salud, Universidad Autónoma Metropolitana, Mexico City 04960, Mexico; asolis@correo.xoc.uam.mx; 2Unidad de Investigación Médica en Farmacología, UMAE Hospital de Especialidades, 2° Piso, Centro Médico Nacional Siglo XXI, Instituto Mexicano del Seguro Social, Av. Cuauhtémoc 330, Col. Doctores, Mexico City 06725, Mexico; valdesguevaramiguel@gmail.com; 3Unidad de Investigación Médica en Genética Humana, UMAE Hospital Pediatría, 2° Piso, Centro Médico Nacional Siglo XXI, Instituto Mexicano del Seguro Social, Av. Cuauhtémoc 330, Col. Doctores, Mexico City 06725, Mexico; romaorr@yahoo.com.mx (R.M.O.-R.); normandgarcia@gmail.com (N.G.-H.); 4Sección de Estudios de Posgrado e Investigación, Escuela Superior de Medicina, Instituto Politécnico Nacional, Salvador Díaz Mirón Esq. Plan de San Luis S/N, Miguel Hidalgo, Casco de Santo Tomas, Mexico City 11340, Mexico; rebc78@yahoo.com.mx

**Keywords:** *Annona muricata*, antitumor potential, cytotoxic activity, 4T1 cells, brine shrimp lethality, acute oral toxicity, breast cancer, flavonoid glycosides, HPLC

## Abstract

*Annona muricata* (**Am**) is a plant used in traditional Mexican medicine to treat cancer. In this study, ethanol extracts of **Am** collected in Acapulco and Tecpan from Guerrero state were evaluated orally on Balb/c mice inoculated with 4T1 cells, for cytotoxic activity (CA) on 4T1 cells, in brine shrimp lethality assay (BSLA), and for acute oral toxicity in mice. In addition, ethanol extracts were subjected to high-performance liquid chromatography (HPLC) with diode array detection. Results showed that the extracts collected in December in Acapulco (AcDe) and Tecpan (TeDe) exhibited the most significant antitumor and cytotoxic activity. In the BSLA, the most important effect was observed in the extracts from Acapulco and Tecpan collected in June (AcJu) and August (TeAg), respectively. The samples from Acapulco (AcJu, and AcAg) and Tecpan (TeJu and TeAg) showed the highest toxicity. The analysis of the extracts, AcDe and TeDe, by HPLC revealed that flavonoids, rutin, narcissin, and nicotinflorin were the major components. These findings suggest that extracts from **Am** collected in Acapulco and Tecpan in the month of December may be an important source to obtain flavonoid glycosides with anticancer potential specifically against breast cancer. This also supports the use of **Am** to treat cancer in Mexican traditional medicine.

## 1. Introduction

Cancer encompasses a large group of diseases characterized by the development of abnormal cells, which divide and grow uncontrollably in any part of the body [1]; it is considered one of the major contributors to premature mortality around the world. It is expected that the global number of cancer patients will increase in the next 50 years due to demographic changes such as population growth and aging [2]. In this sense, breast cancer has been one of the most prevalent and the most common cancers in women around the world, and specifically, in Mexico, with approximately 2.2 million cases in 2020, becoming the leading cause of mortality [3,4]. Chemotherapy, radiotherapy, and surgery are currently essential therapeutic tools used for breast cancer; these treatments are associated with adverse side effects in women as well as their high cost, especially in developing countries. As a result, people diagnosed with cancer are looking to natural products obtained from medicinal plants to complement their treatments. In this context, the scientific community is increasingly interested in more effective, less harmful, and novel natural alternatives.

In Mexico, more than 90% of the population uses medicinal plants to treat their diseases, and between 30 and 70% of cancer patients are willing to use any product obtained from the medicinal plant considering that they perceive these as effective and safe due to their natural origin [5]. Mexico is a country with a great abundance and diversity of natural products that may be a potential source for the discovery of compounds with therapeutic applications [6]. It has been described that about 60% of the drugs approved for cancer have their origin in medicinal plants. There is a large amount of work linked to the search for medicinal plants that have anti-cancer properties, making the natural resources in Mexico an important source for obtaining potential agents for the treatment of cancer [7]. The medicinal uses of the Annonaceae family were reported long ago; it is widely distributed in tropical and subtropical regions. *A. muricata* (Figure 1) has become one of the best known, commercialized, and widely studied species in recent decades due to its therapeutic potential and traditional use [8,9,10].

It is known as “graviola”, “soursop”, and “guanabana” in Mexico is called “zapote de Viejas,” “catucho,” and “cabeza de negro [11].” Their leaves, bark, fruit, seeds, and roots are used for various purposes in traditional medicine for the treatment of cancer, inflammation, skin diseases, analgesic, flu, asthma, colds, malaria, diarrhea, hypertension, and diabetes, among others [12]. Several studies of different extracts obtained from *A. muricata* (fruit, seeds, bark, roots, or leaves) have been reported. In this context, the isolation of roughly 212 secondary active metabolites including acetogenins, alkaloids, polyphenols, and flavonoids have been reported [13,14]. Acetogenins have been identified as one of the major bioactive compounds of *A. muricata* with activity on several cancer cell lines [15,16]. The ethanol extracts obtained from aerial parts from *A. muricata* in combination with doxorubicin showed important cytotoxic activity on 4T1 cells and other cancer cell lines [15,16,17,18,19,20,21,22]. Although there is an extensive amount of information about the cytotoxic properties of *A. muricata*, no information is available regarding its anti-cancer potential in specifically breast cancer using the extracts of plants collected in different periods of the year, and regarding the location of the place of recollection and its influences on their pharmacological and toxic properties. The aim of this study is to evaluate the cytotoxic and antitumor properties using 4T1 cells, BSL, as well as acute oral toxicity of twelve ethanol extracts of **Am** collected for a year in Acapulco (Ac) and Tecpan de Galeana (Te) in Guerrero state, Mexico.

## 2. Results

### 2.1. Yield of the Extracts Obtained of the Leaves from Annona muricata

The yields of ethanolic extracts obtained of leaves from *Annona muricata* (**Am**) are shown in Table 1. The leaves from *Annona muricata* (**Am**) were collected in Acapulco and Tecpan de Galeana, in Guerrero state, Mexico, during the months of February (Fe), April (Ap), June (Ju), August (Ag), October (Oc), and December (De) of 2017. The results showed that best yields of the extract were obtained in April from samples collected in Acapulco and Tecpan de Galeana. In general, the best yields of ethanol extracts were obtained in the samples collected in Tecpan de Galeana, consistent with the characteristic subtropical zone. In contrast, Acapulco is a tropical zone with higher temperatures and higher relative air humidity than Tecpan de Galeana.

### 2.2. Cytotoxicity Assay

The results of cytotoxic activity against 4T1 (Table 2) show that the best effect was observed with the samples collected in Acapulco (AcDe, CC_50_ of 79.2 µg mL^−1^) and Tecpan de Galeana (TeDe, CC_50_ of 75.9 µg mL^−1^). Their activity was closer than doxorubicin (CC_50_ of 62.6 µg mL^−1^), the drug used as a positive control. The remaining extracts showed less activity (CC_50_ > of 87.5 µg mL^−1^). All extracts exhibited a dose-dependent cytotoxic effect (Figure 2). In general, a major effect was observable in the samples from Acapulco.

### 2.3. Anti-Tumor Activity

The evaluation of the antitumor activity (Table 3) of the ethanol extracts from **Am** showed that after the oral administration of the samples, the most active extracts were obtained in December from Acapulco (AcDe, ED_50_ of 10.8 mg kg^−1^) and Tecpan de Galeana (TeDe, ED_50_ of 12.1 mg kg^−1^). The remaining extracts were less active with ED_50_ > 13.3 mg kg^−1^. All extracts exhibited a dose-dependent antitumor effect (Figure 3). In general, a major effect was observable in the samples from Acapulco.

### 2.4. Brine Shrimp Lethality Assay

Results (Table 4) show that the most active ethanol extracts in BSLA were the samples of June and August collected in Acapulco (AcJu, LC_50_ of 1.8 µg mL^−1^) and Tecpan (TeAg, LC_50_ of 2.4 µg mL^−1^), respectively. In general, the extracts obtained from the leaves of **Am** collected in Acapulco showed higher lethality than collected in Tecpan. The remaining extracts showed LC_50_ > 9.7 mg kg^−1^. In general, a major effect was observable in the samples from Acapulco.

### 2.5. Acute Oral Toxicity and Therapeutic Index

The acute oral toxicity was evaluated in agreement with OECD guidelines 423 for the use of natural products in human consumption. The results (Table 5) show that the most toxic samples were those collected in the months of June and August from Acapulco (AcJu and AcAg) and Tecpan (TeJu and TeAg). In contrast, the samples from AcDe and TeDe showed less toxicity. In agreement with acute toxicity and the therapeutic index the most secure extracts were AcDe and TeDe.

### 2.6. HPLC Analysis of the Ethanol Extracts

Analysis of the ethanol extracts, AcDe and TeDe, was performed using high-performance liquid chromatography with diode array detection (HPLC-DAD) and standards of flavonol glycosides, rutin, nicotiflorin, and narcissin. The analysis showed the presence of rutin (32.88 min), nicotiflorin (34.10 min), and narcissin (34.76 min) in AcDe and TeDe (Figure 4). The identification was made by comparing their retention times, UV spectrum (Figure 5), and thin layer chromatography. All extracts showed the presence of rutin, nicotiflorin, and narcissin (Figure 6 and Figure 7).

## 3. Discussion

Breast cancer is a disease present worldwide that affects mainly women aged 20 years. In this context, in 2020, nearly 2.2 million women were diagnosed with breast cancer in the world. The aim of our study was to evaluate the anticancer potential of the ethanol extracts obtained from the leaves of *A. muricata*. The leaves were collected in Acapulco and Tecpan de Galeana from Guerrero state, Mexico in the months of February (AcFe and TeFe), April (AcAp and TeAp), June (AcJu and TeJu), August (AcAg and TeAg), October (AcOc and TeOc), and December (AcDe and TeDe). The anticancer potential was evaluated using in vivo and in vitro assays, including cytotoxic activity on 4T1 cells, the breast cancer model in Balb/c female mice inoculated with 4T1 cells, BSLA, and acute oral toxicity.

In relation to the yields of the ethanol extracts, the results suggest that the geographical location of the cultivated tree may have an influence since the same quantity of plant material was used in the preparation. The samples collected from Tecpan de Galeana, located in a subtropical region, showed the best yields compared to those from Acapulco, located in a tropical region (Figure 1). In this sense, the difference may be due to climatic and environmental factors such as temperature, relative air humidity, the pH of the ground, water precipitation, and radiation [23,24]. Although the difference between the highest and lowest yield is not great, this difference could affect the concentration of active secondary metabolites such as acetogenins and/or flavonoids (Figure 6 and Figure 7) that result in high toxicity or high cytotoxic and antitumor activities.

The next step in our investigation was to evaluate which extracts showed better cytotoxic activity against the breast cancer cell line 4T1 using a WST-1 assay. We observed that all extracts inhibit the proliferation of breast cancer cells as shown in Table 2; our results are consistent with those reported in other studies [20]. In this assay, the CC_50_ values varied markedly between samples. Among these, those collected in December from Acapulco (AcDe) and Tecpan (TeDe) were the most active. Both extracts showed a dose-dependent cytotoxic effect (Figure 2). In addition, the extracts from the Acapulco zone showed the best cytotoxic activity. The results suggest that samples obtained from tropical zones such as Acapulco may have a high concentration of active secondary metabolites such as flavonoid glycosides, alkaloids, and acetogenins. Among these, acetogenins have been reported as the main cytotoxic compounds of *A. muricata* with activity on several cancer cell lines [15,16].

Once we demonstrated the cytotoxic activity of the samples, we decided to evaluate the antitumor activity of the extracts after oral administration, using a breast cancer model in Balb/c female mice inoculated with breast cancer cell line 4T1. The results of the antitumor activity (Table 3) showed a correlation with those obtained in the cytotoxic assay since the most active extracts were obtained in December from Acapulco (AcDe) and Tecpan (TeDe) (Figure 3). In agreement with the cytotoxic effect, in this assay, the extracts, AcDe and TeDe, exhibited dose-dependent antitumor activity.

The results of the antitumor activity showed variations according to its area and time of collection; this can be due to the different concentrations of the metabolites that result in changes in biological activity. In this context, our results agree with other studies that reported biological activity depends on the seasonal variation of *A. muricata* [25]. The secondary metabolites most described that have the ability to decrease tumor growth are acetogenins, alkaloids, and flavonoids [15]. In this sense, acetogenins have been associated with the antitumor and cytotoxic properties of *A. muricata* [15,16,20].

The brine shrimp lethality was performed as an additional assay to evaluate anticancer potential. Meyer et al. [26] introduced this assay with the objective of animal substitution; BS are considered sensitive to a variety of substances, being a useful bioassay for a quick and simple test to predict toxicity and guide the phytochemical fractionation of plant extracts [27,28,29], as well as for the discovery of compounds with cytotoxic and antitumor activity in plant extracts. In order to establish a possible antitumor and cytotoxic potential, it has been reported that in extracts of plants, values of LC_50_ < 100 μg mL^−1^ could be associated with cytotoxic and antitumor activity [29]. Considering the above, our results (Table 4) show that all the extracts obtained exhibited significant lethality, therefore, AcDe and TeDe may be considered important extracts to subject to a bioassay-guided study to obtain secondary metabolites with potential antitumor properties.

Upon demonstrating the in vitro and in vivo antitumor activity of the extracts, the acute oral toxicity was evaluated. All the extracts were evaluated because it was essential to assess if our extracts were a safe treatment; moreover, our results allowed us to determine the toxicity limits and calculate the range of doses that can be used in our next assays in laboratory animals [30]. When the acute toxicity testing was carried out, the animals were observed for 4 h, in accordance with the OCDE 423 guideline, to observe if the administration of the products generated any side effects such as alterations in the nervous system, diarrhea, convulsions, lethargic behavior, tachycardia, or death of the animals. For our investigation, the observation of the animals was important due to an association between the consumption of *A. muricata* and the appearance of atypical Parkinsons [31] and the neurotoxic effects of some of its isolated compounds [13,32]. In this sense, it was necessary to evaluate the extracts and observe the behavior of the animals to discard any of the possible neurotoxic activity that some authors have described. With respect to the toxicity observed in *A. muricata* extracts, it can be influenced by the presence of several metabolites; there are reports which indicate that the most toxic extracts are those enriched with acetogenins [33]. This would lead us to infer that according to our results (Table 5), the extracts collected in Acapulco and Tecpan from June to August may have a higher amount of toxic secondary metabolites such as acetogenins [33]. It is important to point out that, in agreement with the results obtained in acute oral toxicity and antitumor activity, the therapeutic index suggests that the most secure extracts were AcDe and TeDe (Table 5).

Finally, phytochemical analysis helps to know the secondary metabolites present in the plant extracts responsible of its biological activity. In our samples, the HPLC analysis of the ethanol extracts (AcDe and TeDe) from **Am** showed flavonol glycosides as components, which may be related to their cytotoxic, antitumor, and brine shrimp lethality properties including rutin and nicotiflorin. Previous studies have reported the anticancer properties of the flavonol glycoside rutin [34,35]. In this sense, the flavonoids have been reported to interfere in the initiation, promotion, and progression of cancer by modulating different enzymes and receptors in signal transduction pathways related to cellular proliferation, differentiation, apoptosis, angiogenesis, metastasis, and the reversal of multidrug resistance [36]. Additional experiments such as GC-MS and LC-MS analysis must be conducted to know if volatile compounds are present in ethanol extracts from *A. muricata* leaves.

The results of our evaluations showed a correlation between cytotoxic and antitumor properties and brine shrimp lethality to obtain extract candidates with antitumor potential. In addition, it was shown that biological effects depend on the geographical collection area as well as their climatic factors, environmental stress, humidity, rainfall, temperature, soil composition, and prolonged exposure to the sun, among others, which confer on their biological activity [37]. Our study provides important information about the pharmacological use of the extracts of the leaves from *A. muricata* aimed towards antitumoral activity, and exhibits how different variables such as climatic, geographic, etc., can influence their pharmacological activity. Rutin and other flavonoids such as nicotiflorin and narcissin present in **Am** may be responsible for part of the cytotoxic and antitumor activities as well as the brine shrimp lethality activity demonstrated in the ethanol extracts (AcDe and TeDe).

## 4. Materials and Methods

### 4.1. Plant Material

#### 4.1.1. Harvest and Authentication

The leaves of *A. muricata* were collected bimonthly during 2017 in February (Fe), April (Ap), June (Ju), August (Au), October (Oc), and December (De), from two areas in Guerrero, Mexico. The areas selected were Acapulco (16°47′29.6″ N 99°47′28.6″ W) and Tecpan de Galeana (17°15′55.8″ N 100°53′07.6″ W). The specimens were identified by M.Sc Santiago Xolapa Molina, taxonomist from Instituto Mexicano del Seguro Social (IMSS). The corresponding vouchers of specimens (16292a, 16292b, 16292c, 16292d, 16292e, 16292f, 16292g, 16292h, 16292i, 16292j, 16292k, 16292L) are preserved in the institutional Herbarium IMSSM of the IMSS.

#### 4.1.2. Preparation of the Ethanolic Extract of the Leaves from *A. muricata*

The leaves of the plant were cleaned, air-dried, and grounded into a fine powder using an electrical mill, and 500 g of the powdered plant were macerated at room temperature in 6 L of ethanol (EtOH) for a week (twice). The ethanol was filtered and concentrated under reduced pressure to yield a viscous green residue. In total, 12 extracts were obtained and stored at room temperature. They were labelled according to the place and month of collection: Acapulco (Ac) and Tecpan de Galeana (Te), and the first two letters of the month.

### 4.2. Brine Shrimp Lethality Test

The lethality of the ethanolic extracts of *A. muricata* was determined according to the brine shrimp lethality model presented by Meyer, et al. [26], and 10 mg of each extract were dissolved in 1 mL of EtOH. Aliquots were transferred into different tubes to obtain final concentrations of 1000, 100, and 10 μg mL^−1^ in each tube (*n* = 6 for each tube). The EtOH was evaporated then 5 mL of artificial seawater was added, and 10 brine shrimp (BS) were introduced into each tube. After 24 h, surviving BS were counted, and the LC_50_ value was estimated considering the quantity of dead BS at each concentration and using linear regression. LC_50_ values of the extracts < 10 ppm were considered strongly toxic, LC_50_ < 100 ppm moderate, and LC_50_ < 1000 ppm weak, whereas LC_50_ value >1000 μg mL^−1^ was considered non-toxic.

### 4.3. Cell Based Assay

#### 4.3.1. Cell Culture

4T1 murine mammary carcinoma cells were acquired from the American Type Culture Collection (Manassas, VA, USA). Cells were grown in RPMI-1640 medium (biowest) supplemented with 10% fetal bovine serum and 1% penicillin/streptomycin at 37 °C in a humidified atmosphere with 10% CO_2_. Once the cells had attained 80% confluent growth, the cells were trypsinized using Trypsin-EDTA, washed with phosphate buffer (PBS; pH 7.2), and transferred into test tube containing supplemented RPMI-1640 medium, 4T1 cells 1 × 10^5^ cells in 100 µL.

#### 4.3.2. In Vitro Cytotoxicity Assay

Cell cytotoxicity was measured using the WST-1 assay, which is based on the reduction of the tetrazolium salt, WST-1, to formazan by cellular mitochondrial dehydrogenase [38]. Cell proliferation was assessed using the Quick Cell Proliferation Kit II (Abcam, Cambridge, UK, Cat. No. ab65475) according to the manufacturer’s protocol. It consisted of seeding 96-well plates with 4T1 cells at a concentration of 5 × 10^4^ cells per well in a final volume of 100 µL/well of culture medium. After 24 h of incubation, they were treated with different concentrations of the ethanolic extracts of *A. muricata* (50 μg mL^−1^ up to 150 mL^−1^), and each concentration was evaluated in triplicate. Untreated cells were used as negative controls and cells with doxorubicin concentrations (80–150 µM) were used as positive controls. After 24 h, cell viability was evaluated by incubating more cells for 2 h with WST-1 reagent by adding 10 µL to each pellet. At the end of the time, the cells were shaken well for one minute, and cell viability was determined by measuring absorbance at 440 nm, with a microplate reader.

### 4.4. Animal

Healthy, virgin, female Balb/c mice (22 ± 2 g) were used for the in vivo test. They were provided by the animal house of the Centro Medico Nacional, SXXI at Instituto Mexicano del Seguro Social (IMSS) and maintained under controlled temperature and humidity in clean and sterile polyvinyl cages. The animals were manipulated according to the standard specifications for the production, care, and use of laboratory animals of the Mexican Official Norm, NOM-062-ZOO-1999 [39]. All investigations were conducted with the approval of the Specialty Hospital Ethical Committee of Centro Médico Nacional Siglo XXI at IMSS (register: R-2019-3601-024).

#### 4.4.1. Acute Toxicity in Mice

The acute oral toxicity study of the ethanolic extracts of *A. muricata* was conducted in compliance with the OECD (2001) guideline 423 for testing of chemicals, and 111 fasted female Balb/c mice (22 ± 2 g) with free access to water *ad libitum* were randomly assigned as follows: 37 groups of three mice each. One control group was treated with vehicle (2% tween 80 in water), 12 groups were treated with the ethanolic extracts at 30, 12 groups at 300, and 12 groups at 3000 mg kg^−1^. All treatments were dissolved in 2% tween 80 in water with each mouse receiving an amount of 0.5 mL dose orally intragastric route. The signs of toxic effects (convulsions, diarrhea, sleep, etc.) and/or mortality were registered 4 h after the administration, and the animals were observed daily for 14 days. With the results obtained, the median lethal dose (LD_50_) for each extract was calculated. At the end of the experiments, the animals were sacrificed, the internal organs (stomach, gut, kidney, spleen, and liver) were extracted, weighed, and macroscopic observations were performed [40].

#### 4.4.2. In Vivo Anti-Tumor Activity

The mammary tumors were induced with 4T1 cells, which were inoculated subcutaneously in the abdominal mammary gland area in a concentration of 1 × 10^5^ cells/mouse in 100 µL, after 7 days of inoculation. The animals with a palpable tumor were selected [8,41,42]. The animals selected were divided into 38 groups (*n* = 6): negative control (NC), without treatment; positive control (PC), treated with doxorubicin (10 mg kg^−1^ for 8 days); extracts groups (10, 20 and 30 mg kg^−1^). All treatments were administered orally for 8 days. The animals were observed for 28 days from the day of cell inoculation, and the weight and survival of the animals were registered weekly. The evaluation of the antitumor activity was determined by comparing the average weight of the mammary tumor growth in the animals treated with the extract, against animals without treatment.

#### 4.4.3. Characterization of Flavonol Glycosides of the Ethanol Extracts of Leaves from *A. muricata* by High-Performance Liquid Chromatography

The ethanol extracts from **Am** were analyzed using HPLC-diode array detection (DAD) (Waters Agilent, 5301 Stevens Creek Blvd Santa Clara, CA 95051, USA). The analysis was performed using an HPLC-DAD Waters 2795 liquid chromatograph system coupled with a Waters 996 photodiode array detector and an analytical Millennium 3.1 workstation equipped with a C18 analytical column (Waters, Mexico City, Mexico) with dimensions of 250 mm × 4.6 mm and particle size of 5 μm (Spherisorb S50D52, Waters Corporation, Milford, MA, USA). For the analysis, 10 mg of the ethanol extract from leaves from *A. muricata* (**Am**) was disolved in 10 mL of ethanol, and a sample volume of 20 μL from that solution was injected. For elution, a system comprising of a binary mobile phase of acetonitrile solvent/acetic acid 2% in water (A) and acetonitrile 100% (B) was used. The chromatograph operating conditions were programmed to give the following gradients: 1st stage—linear gradient of 80 (A)/20 (B) for 8 min; 2nd stage—linear gradient of 40/60 for 6 min; 3rd stage—linear gradient of 30/70 for 40 min; 4th stage—linear gradient of 90/10 for 6 min with a flow rate of 1 mL min^−1^ of mobile phase. The detection was made at a wavelength (λ) of 254 nm, at room temperature, and a total elution time of 25 min; at the end, the data collected were plotted. Reference standards of rutin, nicotiflorin, and narcissin as well as acetonitrile, ethanol, and acetic acid of HPLC grade were acquired from Sigma and were prepared and analyzed separately under the same conditions described above. In all cases the water used was of HPLC quality, purified in a Milli-Q system (Millipore, Bedford, MA, USA). The presence of substances in the ethanol extracts from leaves of *A. muricata* were confirmed by comparing the retention times and UV spectra with the standard.

#### 4.4.4. Statistical Analysis

The results are expressed as mean ± standard error of the mean (SEM). Statistical analysis of the data was performed using one-way ANOVA and multiple comparison tests using Dunnett with a value of *p* < 0.05 to establish a significant difference between the study groups. The EC_50_, LD_50_, and LC_50_ were calculated by linear interpolation of the percentage mortality values for each concentration. All analysis was performed using Graph Pad Prism version 8.

## 5. Conclusions

This study proposes the analysis of 12 extracts using in vivo and in vitro assays including cytotoxic activity on 4T1 cells, breast cancer model in Balb/c female mice inoculated with 4T1 cells, BSLA, and acute oral toxicity. In addition, some aspects of the relationship between their collection and their biological activities were demonstrated. In this sense, the careful selection of the leaves from **Am** in relation to growing area and season is imperative, as both determine its antitumor activity. In addition, the present study suggests that the **Am** plant could be a material with potential anticancer properties that could be of great importance for tumor inhibition in breast cancer. Further experiments are needed to gain knowledge about its antitumor activity and mechanisms and to continue with bio-guided fractionation to isolate the secondary metabolites responsible for the cytotoxic and antitumor properties and brine shrimp lethality observed to ethanol extracts (AcDe and TeDe) from **Am**.

## Figures and Tables

**Figure 1 molecules-26-07675-f001:**
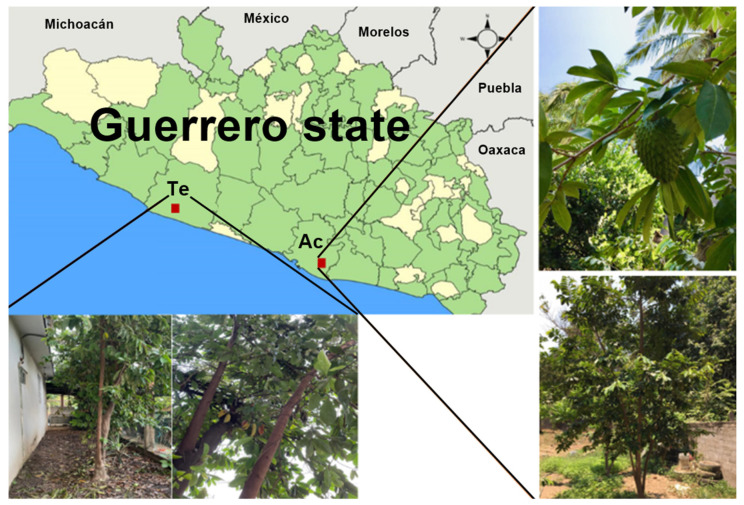
Map of Guerrero state showing Acapulco (**Ac**) and Tecpan de Galeana (**Te**) regions where the leaves of *A. muricata* were collected.

**Figure 2 molecules-26-07675-f002:**
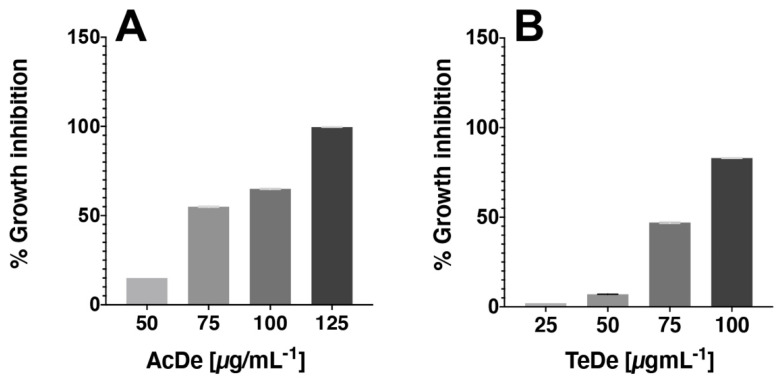
Representative WST-1 assay showing the cytotoxicity activity of AcDe (**A**) and TeDe (**B**) in 4T1 type of cancer cell after 24 h of incubation in vitro.

**Figure 3 molecules-26-07675-f003:**
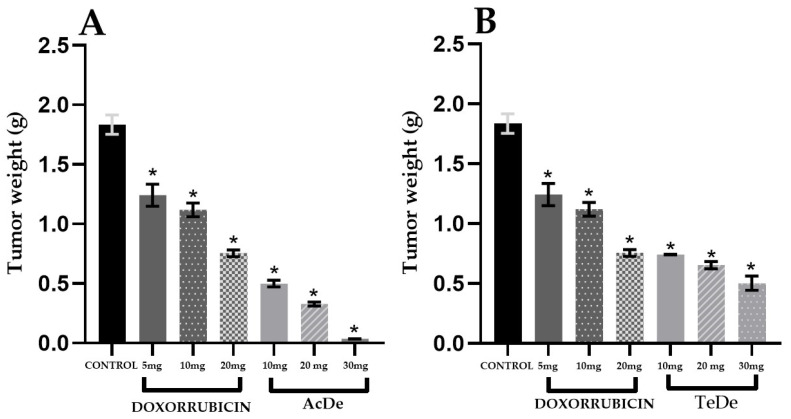
In vivo tumor 4T1 tumor bearing mice growth inhibition experiment (*n* = 6). Control: tumor weight harvested from untreated group (control), Doxorubicin: pharmacological control. (**A**) Tumor weight of group treated with different doses of AcDe; (**B**) Tumor weight of group treated with different doses of TeDe. Each value represents the mean ± standard error of the mean. * *p* < 0.05 vs. Control group.

**Figure 4 molecules-26-07675-f004:**
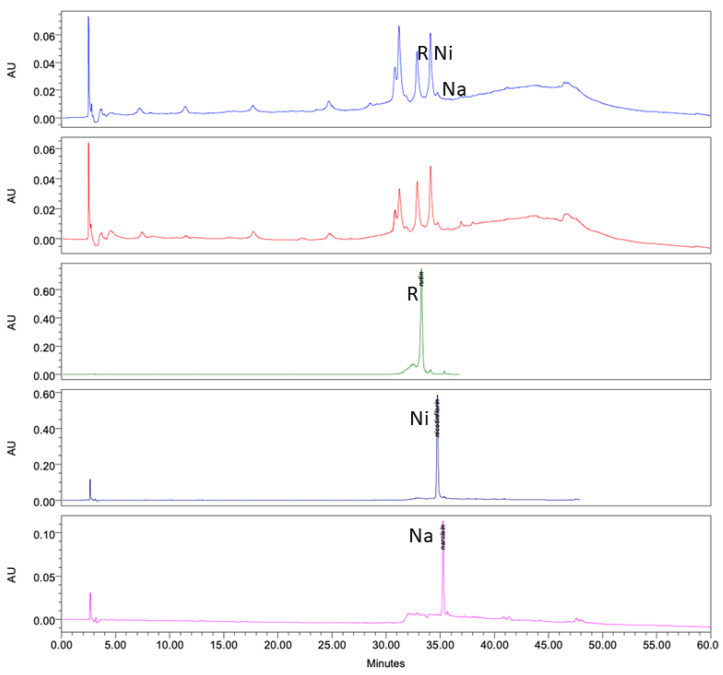
High-performance liquid chromatography with diode-array detection (HPLC-DAD) analysis at 254 nm of ethanol extract from *A. muricata* leaves. TeDe (blue) and AcDe (red); flavonol glycosides standards, rutin (R, green), nicotiflorin (Ni, gray), and narcissin (Na, magenta).

**Figure 5 molecules-26-07675-f005:**
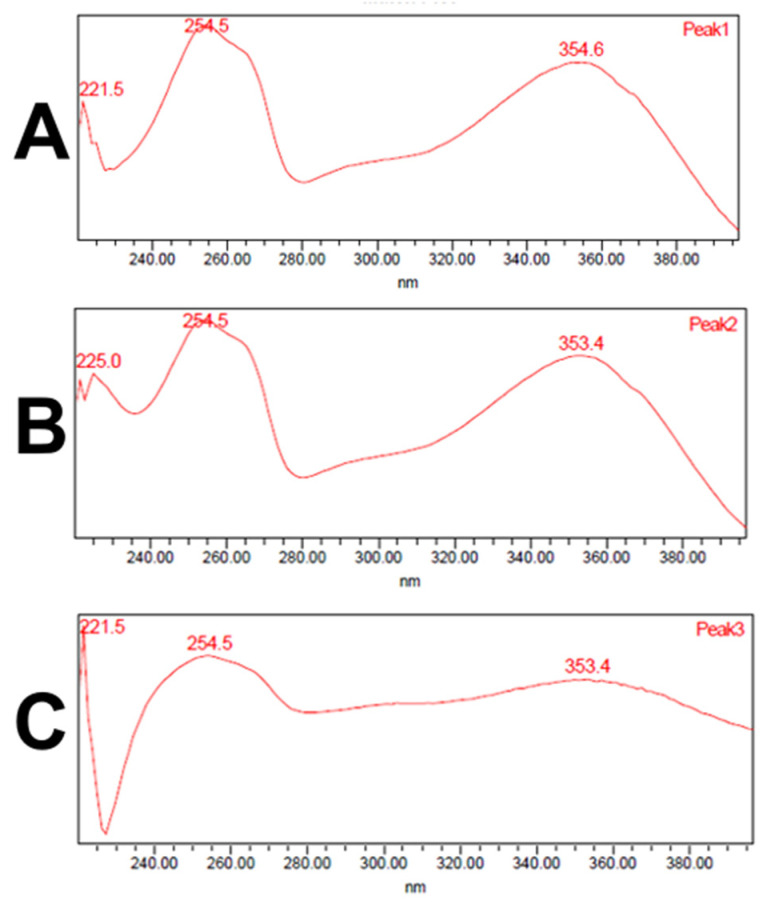
UV spectrum obtained from HPLC-DAD analysis of rutin (**A**), nicotiflorin (**B**), and narcissin (**C**).

**Figure 6 molecules-26-07675-f006:**
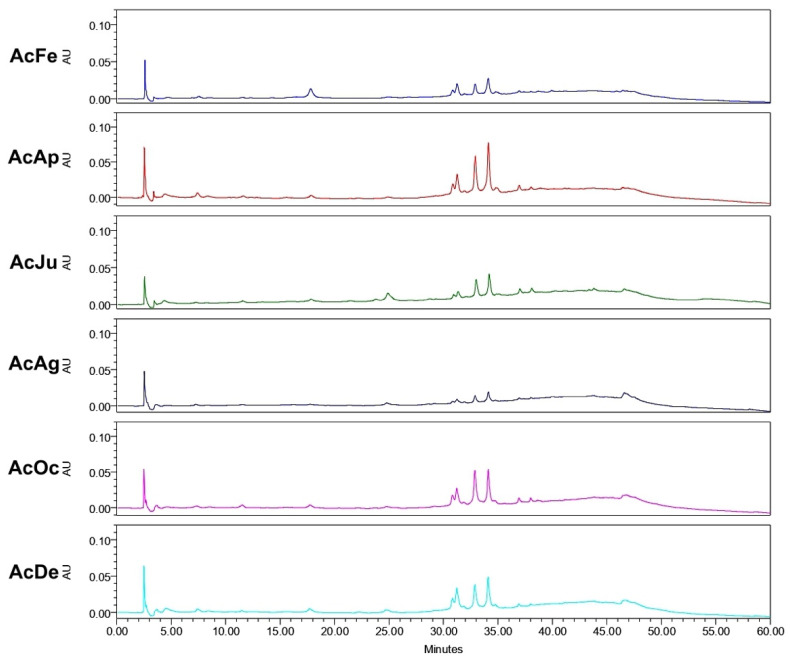
High-performance liquid chromatography with diode-array detection (HPLC-DAD) analysis of ethanol extracts from *A. muricata* leaves collected in Acapulco, Mexico.

**Figure 7 molecules-26-07675-f007:**
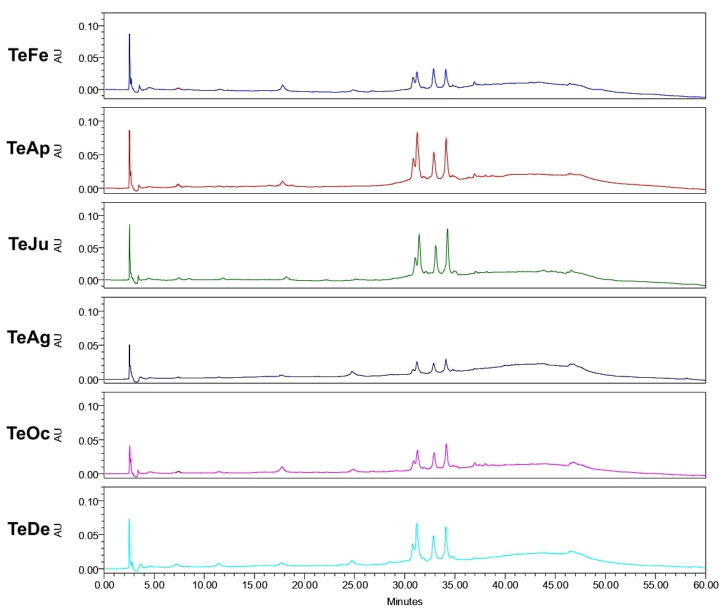
High-performance liquid chromatography with diode-array detection (HPLC-DAD) analysis of ethanol extracts from *A. muricata* leaves collected in Tecpan de Galeana, Mexico.

**Table 1 molecules-26-07675-t001:** Yield of extracts obtained of leaves from *Annona muricata* collected in Acapulco and Tecpan de Galeana in Guerrero, Mexico for one year *.

Zone/Month	Extract ID	Fresh Weight of Leaves (g)	Ethanolic Extract Obtained (g)	Yield (%)
Acapulco/February	AcFe	500	52.1	10.4
Acapulco/April	AcAp	500	62.4	12.5
Acapulco/June	AcJu	500	40.1	8.0
Acapulco/August	AcAu	500	42.4	8.5
Acapulco/October	AcOc	500	48.7	9.7
Acapulco/December	AcDe	500	48.0	9.6
Tecpan/February	TeFe	500	50.5	10.1
Tecpan/April	TeAp	500	73.1	14.6
Tecpan/June	TeJu	500	71.3	14.3
Tecpan/August	TeAu	500	42.4	8.5
Tecpan/October	TeOc	500	65.0	13.0
Tecpan/December	TeDe	500	50.0	10.0

* All plant material collection was carried out on 2017.

**Table 2 molecules-26-07675-t002:** Cytotoxic activity of twelve ethanol extracts obtained of the leaves from *Annona muricata* in CC_50_ (µg mL^−1^) against 4T1 cells.

Treatment	CC_50_ (µg mL^−1^)	Treatment	CC_50_ (µg mL^−1^)
AcFe	139.1 ± 2.2	TeFe	128.1 ± 0.4
AcAp	99.9 ± 0.2	TeAp	97.5 ± 0.3
AcJu	87.4 ± 0.5	TeJu	98.7 ± 0.1
AcAg	122.5 ± 0.3	TeAg	126. 3 ± 0.2
AcOc	89.1 ± 0.2	TeOc	133.7 ± 0.4
AcDe	79.2 ± 0.2	TeDe	75.9 ± 0.1
Doxorrubicin	62.6 ± 0.8		62.6 ± 0.8

Data are expressed as mean ± S. E. M. (*n* = 3); CC_50_: cytotoxic concentration 50.

**Table 3 molecules-26-07675-t003:** Results of antitumor activity of the ethanol extracts from *A. muricata* leaves in a breast cancer model with 4T1 cells.

Treatment	ED_50_ (mg kg^−1^)	Treatment	ED_50_ (mg kg^−1^)
AcFe	13.3 ± 0.06	TeFe	16.2 ± 0.42
AcAp	20.8 ± 0.49	TeAp	30.3 ± 2.18
AcJu	24.8 ± 0.20	TeJu	16.9 ± 1.06
AcAg	40.6 ± 1.9	TeAg	20.7 ± 0.22
AcOc	15.6 ± 0.32	TeOc	14.4 ± 0.46
AcDe	10.8 ± 0.014	TeDe	12.1 ± 0.19
Doxorrubicin	15.57 ± 0.76		15.57 ± 0.76

Data are expressed as mean ± S. E. M. (*n* = 6); ED_50_: effective doses 50.

**Table 4 molecules-26-07675-t004:** Brine shrimp lethality of the extracts from *A. muricata* leaves.

Treatment	LC_50_ (µg mL^−1^)	Treatment	LC_50_ (µg mL^−1^)
AcFe	44.7 ± 0.89	TeFe	60.5 ± 3.36
AcAp	47.3 ± 0.06	TeAp	40.5 ± 0.29
AcJu	1.8 ± 0.34	TeJu	39.6 ± 1.91
AcAg	9.7 ± 0.80	TeAg	2.4 ± 0.62
AcOc	23.4 ± 1.50	TeOc	36.2 ± 5.39
AcDe	22.9 ± 1.50	TeDe	50.6 ± 0.32
Doxorrubicin	>500		>500

Data are expressed as mean LC_50_ ± S. E. M. (*n* = 3).

**Table 5 molecules-26-07675-t005:** Acute oral toxicity and therapeutic index of the extracts from *A. muricata* leaves.

Treatment	LD_50_ (mg kg^−1^)	TI^a^	Treatment	LD_50_ (mg kg^−1^)	TI ^a^
AcFe	1585	119.2	TeFe	1585	97.8
AcAp	1585	76.2	TeAp	1585	52.5
AcJu	165	6.7	TeJu	232	13.7
AcAg	165	4.1	TeAg	232	11.2
AcOc	1515	97.1	TeOc	1585	110.1
AcDe	1515	140.7	TeDe	1585	131.0

Correlation coefficient > 0.9500; Data are expressed as mean ± S. E. M. (*n* = 3); LD_50_: Lethal dose 50. ^a^ TI: therapeutic index calculated as LD_50_ (acute oral toxicity)/ED_50_ (antitumor activity).

## Data Availability

The data presented or additional data in this study are available on request from the corresponding author.

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
