# Peer review of "Antitumor Potential of Annona muricata Linn. An Edible and Medicinal Plant in Mexico: In Vitro, In Vivo, and Toxicological Studies"

_molecules, 2021, doi:10.3390/molecules26247675_

Round 1

Reviewer 1 Report

The authors have evaluated several collection of a traditional medicine, Annona muricata, for antitumor and other types of activities. The topic is worthwhile and the paper is interesting. However, it appears the authors have not interpreted all of the data in a comprehensive manner. Also, the paper is rather disjointed.

Some comments follow.

Table 1. Although the yield of the extracts do vary, this does not seem to be a meaningful finding. The difference between the highest and lowest yield is not that great. Moreover, the qualitative aspects may be more important than the quantitative aspects. Augmentation of the discussion is suggested.

Table 2. In the text, extracts are characterized as inactive with values of >87µg/ml, but 87 is less than some of the values given in the table.  Also, the IC50 values are relatively high, but it should be noted that the incubation period was only 24 h. IC50 values tend to decrease with increasing incubation times. This should be noted.

Table 3. It is odd that antitumor data is shown as a figure in the discussion and a more comprehensive presentation is given in the table. Apparently, all of the extracts were evaluated in the in vivo model. The emphasize placed on the December collection seems to be displaced since all of the extracts demonstrated discernable activity. This should be discussed.

Table 4. Having the ability to perform studies with cultured mammalian cells and tumors in mice is sufficient. The impact of these data is diluted by the inclusion of brine shrimp data, which are rather trivial and more indicative of toxicity than efficacy. It is recommended that brine shrimp data be removed from the paper throughout.

The introduction states 2.2 million cases of breast cancer and the discussion states 2.3 million.

It is odd that results are presented in the discussion section rather than the results section. This needs to be coordinated. The results should be in the results section.

Given the LD50 values (Table 5) and the ED50 values (Table 3), it is surprising the authors did not discuss therapeutic indices. These indices may be approximate but should be favorable. Comparison can be made with doxorubicin, since it was used as a control (perhaps derived from the literature).

Is in important to note in the discussion, abstract, etc., that the test material was administered orally for the antitumor study. Also, the authors state that body weight was determined weekly, but these data are not given. This is important.

The paper is generally well-written and intelligible. However, editing is required. For example, the Introduction could benefit my more paragraph breaks, and there are several misspelled words. Actually, there are many misspelled words throughout.

Author Response

Mrs. Emilia Kuzniak-Glanowska

Assistant Editor, MDPI Krakow

ANSWER TO EDITOR AND REVIEWER

In agree with editor and referees comments we decided revise the manuscript “Antitumor Potential of Annona muricata Linn. an Edible and Medicinal
Plant in Mexico: In Vitro, In Vivo, and Toxicological Studies” (molecules-1502195).

all changes made are in yellow color, including:

Comments and suggestion for Authors

Reviewer 1

Query1

Table 1. Although the yield of the extracts do vary, this does not seem to be a meaningfull finding. The difference between the highest and lowest yield is not great. Moreover, the qualitative aspects may be more important than the quantitative aspects.  Augmentation of the discussion is suggested.

Answer:

A short phrase was included:

Although the difference between the highest and lowest yield is not great this difference could affect in the concentration of active secondary metabolites such as acetogenins that result in high toxicity or high cytotoxic and antitumor activities.    

Query2

Table 2. In the text, are characterized as inactive with values of >87 mg/ml but 87 is less than some of the values given in the table. Also, the IC50 values are relatively high, but is should be noted that the incubation period was only 24h. IC50 values tend to decrease with increasing incubation times. This should be noted.

Answer:

Additional phrase included:

their activity was close than doxorubicin (CC50 of 62.6 µg/ml) drug used as positive control. The remaining extracts showed less activity CC50 > of 87.5 µg/ml

Query3

Table 3. It is odd that antitumor data is shown as a figure in the discussion and a more compressive presentation is give in the table. Apparently, all of the extracts were evaluated I the in vivo model. The emphasize placed on the December collection seems to be displaced since all of the extracts demonstrated discernable activity. This should be discussed.

Answer:

In agreement with suggestion of referee the figures were changed to result section.

Query4

Having the ability to perform studies with cultured mammalian cells and tumor in mice is sufficient. The impact of these data is diluted by inclusion of brine shrimp data be removed from the paper throughout.

Answer:

With the propose of don’t use animals and cells in the first phases in other future research the know BSL assay was evaluated it is easy and cheap and restrain the use of animals. We found correlation with cytotoxic and antitumor activities therefore decided conserve this part in this manuscript. 

Query5.

The introduction states 2,2 million cases of breast cancer and the discussion states 2.3 million

Answer:

Number was corrected

Query6

It is odd that results are presented in the discussion rather than the results section. This needs to be coordinated. The results should be in the results section.

Answer:

The figures were changed in the results section

Query7

Given the LD50 values (Table 5) and the ED50 value (Table 3), it is surprising the authors did not discuss therapeutic indices. These indices may be approximate but should be favorable. Comparison can be made with doxorubicin, since it was used as a control (perhaps derived from the literature).

Answer:

In agreement with OCDE 423 (drugs for human use) with this date is enough to know if a product is or not toxic. In this sense an extensive discussion was subject in this part. However, in agreement with the referee therapeutic indexes were calculated. See Table 5. Also, additional phrase was included in the Discussion.

Query8

Is in important to note in the discussion, abstract, etc. that the test material was administered orally for the antitumor study. Also, the authors state that body weight was determined weekly, but these data are not given. This is important.

Answer:

In agreement with the suggestion of referee in the abstract, result, discussion and methods was indicated that materials were administrated orally.

In relation of body weight these don’t exhibit significant changes for this reason don’t was included.

Query9

The paper is generally well-written and intelligible. However, editing is required. For example, the Introduction could benefit my more paragraph breaks, and there are several misspelled words. Actually, there are many misspelled words throughout.

Answer:

Considering that referee 3 suggest that manuscript “manuscript deserves to be published as it is”, referee 2 suggest minor English changes are required, and referee 1 said “The paper is generally well-written and intelligible”.

 All manuscript was reviewed in spelling for all authors. In special introduction was benefit. To obtain a best manuscript.

Dr. Fernando Calzada

Reviewer 2 Report

Dear Authors,

The manuscript presents interesting results, mainly considering the wide interest in natural products with anticancer properties. However, should be improved.

1) there is plenty of information about A. muricata compounds and anticancer activity, mainly breast cancer. However, you uperficially commented about it and in the discussion, no mention is done on acetogenins

2) you wrote: "In this study the HPLC analysis of the ethanol extracts (AcDe and TeDe) from Am showed flavonol glycosides as main components,
which may be related to their cytotoxic, antitumor, and brine shrimp lethality properties". Considering HPLC-DAD detects only compounds able to be detected by UV,  it is not possible to affirm those compounds are the main components, based on only that technique.

3) you should present not only the HPLC chromatogram but also the UV spectrum for those detached peaks.

4) you should present the HPLC-DAD analysis for all evaluated extracts. 

Author Response

Mrs. Emilia Kuzniak-Glanowska

Assistant Editor, MDPI Krakow

ANSWER TO EDITOR AND REVIEWER

In agree with editor and referees comments we decided revise the manuscript “Antitumor Potential of Annona muricata Linn. an Edible and Medicinal
Plant in Mexico: In Vitro, In Vivo, and Toxicological Studies” (molecules-1502195).

all changes made are in yellow color, including:

Comments and suggestion for Authors

Reviewer 2

Query1

There is plenty of information about A. muricata compounds and anticancer activity, mainly breast cancer. However, you superficially comment about it and in the discussion, no mention is done on acetogenins.

Answer:

Additional phrases were included in the section of Discussion on acetogenins as antitumor and cytotoxic compounds of A. muricata.

Page 6 and 7 respectively:

, alkaloids, and acetogenins. Among these, acetogenins have been reported as the main cytotoxic compounds of A. muricata with activity on several cancer cell lines [15,16].

In this sense, acetogenins have been associated with the antitumor and cytotoxic properties of A. muricata [15, 16, 20].

Query2

You wrote “In this study the HPLC analysis of the ethanol extracts (AcDe and TeDe) from Am showed flavonol glycosides as main components. Which may be related to their cytotoxic, antitumor, and brine shrimp lethality properties”.

Considering HPLC-DAD detects only compounds able to be detected by UV, it is not possible to affirm those compounds are the main components, based on only that technique.

Answer

The phrases of this part were restructured as:

In our samples, the HPLC analysis of the ethanol extracts (AcDe and TeDe) from Am showed flavonol glycosides as components, which may be related to their cytotoxic, antitumor, and brine shrimp lethality properties including rutin and nicotiflorin. Previous studies have been reported the anticancer properties of the flavonol glycoside rutin [34, 35]. In this sense the flavonoids have been reported to interfere in the initiation, promotion, and progression of cancer by modulating different enzymes and receptors in signal transduction pathways related to cellular proliferation, differentiation, apoptosis, angiogenesis, metastasis and reversal of multidrug resistance [36]. Additional experiment such as GC-MS and LC-MS analysis must be realized to know volatile compounds present in ethanol extracts of A. muricata leaves.

Query3

You should present not only the HPLC chromatogram but also the UV spectrum for those detached peaks.

Answer:

UV spectrum were included

Query4

You should present the HPLC-DAD analysis for all evaluated extracts.

Answer:

HPLC-DAD of all extracts today are including.

Dr. Fernando Calzada

Reviewer 3 Report

This manuscript deserves to be published as it is

Author Response

Mrs. Emilia Kuzniak-Glanowska

Assistant Editor, MDPI Krakow

ANSWER TO EDITOR AND REVIEWER

In agree with editor and referees comments we decided revise the manuscript “Antitumor Potential of Annona muricata Linn. an Edible and Medicinal
Plant in Mexico: In Vitro, In Vivo, and Toxicological Studies” (molecules-1502195).

all changes made are in yellow color, including:

Comments and suggestion for Authors

Reviewer 3

Query 1.

No query was suggested

Referee 3 suggest. This manuscript deserves to be published as it is

Dr. Fernando Calzada
